# The Association between Childhood Exposure to Ambient Air Pollution and Obesity: A Systematic Review and Meta-Analysis

**DOI:** 10.3390/ijerph19084491

**Published:** 2022-04-08

**Authors:** Chao Huang, Cheng Li, Fengyi Zhao, Jing Zhu, Shaokang Wang, Guiju Sun

**Affiliations:** 1Key Laboratory of Environmental Medicine and Engineering of Ministry of Education, Department of Nutrition and Food Hygiene, School of Public Health, Southeast University, 87 Dingjiaqiao Road, Nanjing 210009, China; 220203874@seu.edu.cn (C.H.); zfy3319@foxmail.com (F.Z.); gjsun@seu.edu.cn (G.S.); 2Beijing Institute of Nutritional Resources, Beijing 100069, China; chengli_nutri@163.com (C.L.); jingzhu.nutri@outlook.com (J.Z.)

**Keywords:** childhood, air pollution, obesity, BMI, meta-analysis

## Abstract

Obesity has become a worldwide epidemic; 340 million of children and adolescents were overweight or obese in 2016, and this number continues to grow at a rapid rate. Epidemiological research has suggested that air pollution affects childhood obesity and weight status, but the current evidence remains inconsistent. Therefore, the aim of this meta-analysis was to estimate the effects of childhood exposure to air pollutants on weight. A total of four databases (PubMed, Web of Science, Embase, and Cochrane Library) were searched for publications up to December 31, 2021, and finally 15 studies met the inclusion criteria for meta-analysis. Merged odds ratios (ORs), coefficients (β), and 95% confidence intervals (95% CIs) that were related to air pollutants were estimated using a random-effects model. The meta-analysis indicated that air pollutants were correlated with childhood obesity and weight gain. For obesity, the association was considerable for PM_10_ (OR = 1.12, 95% CI: 1.06, 1.18), PM_2.5_ (OR = 1.28, 95% CI: 1.13, 1.45), PM_1_ (OR = 1.41, 95% CI: 1.30, 1.53), and NO_2_ (OR = 1.11, 95% CI: 1.06, 1.18). Similarly, BMI status increased by 0.08 (0.03–0.12), 0.11 (0.05–0.17), and 0.03 (0.01–0.04) kg/m^2^ with 10 μg/m^3^ increment in exposure to PM_10_, PM_2.5_, and NO_2_. In summary, air pollution can be regarded as a probable risk factor for the weight status of children and adolescents. The next step is to conduct longer-term and large-scale studies on different population subgroups, exposure concentrations, and pollutant combinations to provide detailed evidence. Meanwhile, integrated management of air pollution is essential.

## 1. Introduction

Obesity has become a worldwide epidemic and urgent health issue [1]. The prevalence of overweight and obesity has increased considerably in the last few decades and nearly tripled since 1975. With regard to children and adolescents, 340 million of them were overweight or obese in 2016, and this number continues to grow at a rapid rate [2,3]. Childhood obesity has been associated with obesity and increased risks for chronic disease in adulthood [4,5], and these adverse effects may last the whole lifetime [6]. Research has identified multiple factors that can lead to childhood obesity and it has been shown that childhood obesity can be attributed to genetic, dietary, and behavioral factors [7]. Despite genetic and metabolic predispositions, the rising epidemic of obesity indicates environmental factors may play a role in accelerating the progression of childhood obesity [8].

The Global Burden of Disease study revealed that air pollution can be the most adverse environmental health hazard for disease and mortality worldwide [9,10]. Well over 80% of urban dwellers suffer from air pollution, and the most seriously affected individuals were low-income residents [11]. In recent years, mounting evidence suggests that air pollution can be an obesogenic factor [12]. It is mainly through the biochemical and behavioral pathways that air pollution affects body weight. Metabolic disorders [13], inflammatory reactions [14], reduced sleep duration and quality [15] that are caused by air pollution all contribute to the accumulation of adipose tissue and weight gain, as has been demonstrated in animal trials. Additionally, the decline in air quality reduces a person’s willingness to engage in outdoor activities [16], which in turn increases indoor time, in order to reduce the impact of pollution on the human body [17].

Although there are numerous original studies on air pollution and obesity, the effects remained inconsistent and differed among the populations, pollutant types, and pollutant concentrations [12]. We found that previous studies were more concentrated on exposure during pregnancy [18,19], whereas recent studies have increasingly examined the direct effects of children’s exposure to air pollution on obesity [20,21]. However, the findings seem to be inconsistent even in studies of children and adolescents only [22,23,24]. For instance, Fioravanti et al. suggested that the evidence that air pollution causes obesity was limited [23]. In contrast, some studies indicated that long-term exposure to air pollutants might be correlated with weight gain and the development of obesity [20,21,25,26]. Among the available review articles, the existing studies have been mostly concentrated on adults or whole populations, and quantitative synthesis of the contribution of air pollution to children and adolescents remains scarce [12,27,28]. A meta-analysis by Parasin et al. examined the relationship between air pollution and childhood obesity, but did not distinguish between exposure during pregnancy and individual exposure, and also did not standardize when combining the effects of pollutants across studies [29]. Therefore, we believe that the topic still has potential for further research. As such, that this study aimed to systematically review and quantitatively analyze the scientific evidence on the influence of exposure to air pollution on weight gain and obesity in childhood.

## 2. Methods

### 2.1. Search Strategy

The systematic review and meta-analysis were based on the Preferred Reporting Items for Systematic Review and Meta-Analysis (PRISMA) guidelines [30]. A literature search was conducted through PubMed, Web of Science, Embase, and Cochrane Library to examine the relationship between childhood exposure to air pollutants and weight gain. The keywords included a combination of three main aspects, which were used to represent exposure (air pollutants), outcome (body weight status), and population (children and adolescents). The search strategies are presented in the Appendix A. When searched in PubMed and Embase, the “[All fields]” tag was used. The search function “TS = Topic” was applied in Web of Science, represents topic term search limited to the fields of title, abstract, keyword, and Keyword Plus [31]. No restrictions were placed on the start time in the window of the search, for the period up to and including 31 December 2021, but the language was limited to articles in English. We also conducted a backward reference search and forward reference search based on the full-text articles meeting the study selection criteria in the search strategy while no additional studies were found.

### 2.2. Eligibility Criteria and Study Selection

The inclusion criteria were based on the following principles: (1) population: conducted on children and adolescents (≤18 years old); (2) exposure: short-term (<3 months) or long-term (≥3 months) exposure to ambient air pollution (PM, NO_x_, SO_x_, CO, O_3_); (3) outcome: overweight or obesity status measured by body mass index (BMI), waist circumference (WC), waist-to-height ratio (WHtR), skinfold thickness, or body fat; (4) article type: original research; and (5) article language: written in English. For articles to be include in the meta-analysis, the outcome indicators of interest were further restricted to provide the relative risks (RRs)/odd ratios (ORs)/hazard ratios (HRs)/coefficients (β), and corresponding 95% confidence intervals (CIs). Studies were excluded from the review if they met any of the criteria below: (1) studies conducted in adults only; (2) prenatal exposure; (3) body weight status includes birth weight only; (4) animal experimental studies; (5) studies on the effects of passive smoking, wood smoke, and (environmental) endocrine-disrupting chemicals; and (6) letters, editorials, protocols, or review articles.

### 2.3. Data Extraction and Quality Assessment

For the eligible studies, the extracted data included: basic information of studies (authors, publication year, country, study database, study design, study period, sample size, age, gender proportion), and exposure and outcome indicators (exposure type, exposure assessment, statistical model, adjusted covariates, effect estimates).

The National Institutes of Health’s Quality Assessment Tool for Observational Cohort and Cross-Sectional Studies [32] was used to assess the quality of the included research in the meta-analysis. The scale assessed each study in terms of 14 criteria, covering several aspects of study objectives, sample selection, exposure and outcome measurement, and statistical analysis. The total score ranged from 0 to 14 and was calculated by adding up the scores for each criterion. A quality assessment of studies was used to assist in measuring the strength of scientific evidence, but not for determining the inclusion and exclusion of studies.

The selection of studies, data extraction, and quality assessment were carried out independently by two reviewers, with disagreements resolved by a third reviewer.

### 2.4. Data Synthesis and Statistical Analysis

Weight and height were measured by professionals according to clinical standard protocols. Overweight and obesity were then defined according to different regional standards. Long-term exposure means exposure for longer than 3 months [28].

In this review, a random-effects model was used to assess the combined effects and the 95% CIs by incorporating RRs/ORs/HRs for binary outcome (obesity) and β for continuous outcome (BMI) from initial studies [28]. Since the definition of pollutant concentration increments varied across studies, we defined 10 μg/m^3^ as the standard increment; other reported units were converted (formulas: NO_2_: 1 ppb = 46/22.4 μg/m^3^; NO_x_: 1 ppb = 46/22.4 μg/m^3^; O_3_: 1 ppb = 48/22.4 μg/m^3^). Based on the assumption of a linear relationship between air pollution and obesity or BMI, the following equations were used to standardize the estimates of effects across studies [28,33]:OR_(standardized)_ = OR_(original)_ ^Increment(10)/Increment(original)^(1)
β_(standardized)_ = β_(original)_ × Increment(10)/Increment(original)(2)

In addition, when multiple models were available in the study or estimates from sensitivity analyses were reported, we used the full-adjusted model only, or the main model that was indicated by the researchers. For studies with different groups (such as gender, age group) for which overall effects were not accessible, we treated them as a separate research based on their respective sample sizes [21,22,34].

In order to examine the heterogeneity of the included studies, we used Chi-squared test and *I*^2^ statistics, either *I*^2^ > 50% or *p*-value of Chi-squared test < 0.10 was considered as statistically significant heterogeneity [35]. Publication bias was evaluated using Egger’s test and funnel plot [36]. We also conducted subgroup analyses to explore sources of heterogeneity according to study design (cohort or cross-sectional), country (China or others), and study quality (<13 or ≥13). To ascertain whether a particular study had an undue influence upon the overall results, a sensitivity analysis was carried out using the leave-one-out method. All statistical analyses were conducted using STATA (version 13.1). Statistically significant differences were determined as two-tailed *p*-value < 0.05.

## 3. Results

### 3.1. Study Selection

Figure 1 shows the flowchart of study selection. The initial keyword search identified 6488 records, including 1699 from PubMed, 2229 from Web of Since, 2491 from Embase, and 69 from Cochrane Library. After the removal of duplicate records, 4892 unique titles and abstracts were assessed, and 4871 records were further excluded. The full texts of the remaining 21 articles were reviewed, 20 met the inclusion criteria (one author manuscript excluded). Among the 20 studies that met the review inclusion criteria, 5 studies were further excluded due to inconsistency with the intended required outcome effects [24,37,38,39], and unavailability exposure dose [8]. A total of 15 studies were finally included in the meta-analysis [20,21,22,23,26,34,40,41,42,43,44,45,46,47,48].

### 3.2. Study Characteristics

The characteristics of the included studies are presented in Table 1. These studies were carried out in seven countries: six in China, three in Spain, two in the United States, and one in each of Italy, Mexico, the Netherlands, and the United Kingdom. Overall, the data reported on 683,081 participants; all the subjects were children and adolescents with two studies only for children under the age of five. As for study design, eight were cross-sectional studies while seven were cohort. All studies were long-term (≥3 months) exposures. In terms of research quality assessment, eight achieved a score of 13 and were considered as good quality (Appendix A).

Table 2 demonstrates the exposure, outcome information, and association assessment methods. In order to assess the exposure to air pollutants, thirteen studies used model estimation (satellite-based spatial-temporal model, hybrid spatio-temporal model, land use regression model, machine-learning model, California line-source dispersion model) while two used monitoring station data directly. Weight status and obesity were mainly based on BMI and its related indicators (except for one which used waist circumference), meanwhile the reference standard for eight studies was based on international standards and seven on national standards.

### 3.3. Air Pollution on Obesity and BMI in Children and Adolescents

The association of childhood obesity and air pollutants was estimated using a pooled ORs, respectively 9, 11, 3, 2, and 11 studies investigated the effects of obesity in relation to PM_10_, PM_2.5_, PM_1_, O_3_, and NO_2_ exposure (Table 3). The meta-analysis results (Appendix A) showed that long-term exposure to air pollution could increase the risk of childhood obesity (Figure 2), the only pollutant that exhibited no significant correlation was O_3_ (OR = 1.08, 95% CI: 0.99,1.18), while the association was considerable for PM_10_ (OR = 1.12, 95% CI: 1.06,1.18), PM_2.5_ (OR = 1.28, 95% CI: 1.13,1.45), PM_1_ (OR = 1.41, 95% CI: 1.30,1.53), and NO_2_ (OR = 1.11, 95% CI: 1.06,1.18).

The relationships between PM_10_, PM_2.5_, NO_2_, and NO_x_ and childhood BMI were reported by 3, 3, 5, and 2 studies (Table 3). The BMI status increased by 0.08 (0.03–0.12), 0.11 (0.05–0.17), and 0.03 (0.01–0.04) kg/m^2^ with 10 μg/m^3^ increment in exposure to PM_10_, PM_2.5_, and NO_2_, respectively. Exposure to NO_x_, however, was not significantly associated with BMI growth (Figure 3).

### 3.4. Heterogeneity, Publication Bias, and Sensitivity Analysis

In studies with obesity as an outcome, heterogeneity existed in the analysis of air pollutants (PM_10_: *I*^2^ = 85.9, *p* < 0.001; PM_2.5_: *I*^2^= 86.3, *p* < 0.001; O_3_: *I*^2^ = 71.5, *p* = 0.061; NO_2_: *I*^2^ = 84.1, *p* < 0.001), except for PM_1_ (*I*^2^ = 0, *p* = 0.905). Funnel plots (Appendix A) and an Egger’s test showed potential publication bias that was only identified in NO_2_ (PM_10_: *p* = 0.076; PM_2.5_: *p* = 0.238; PM_1_: *p* = 0.324; NO_2_: *p* = 0.001; O_3_: N/A). 

For the outcome of BMI status, heterogeneity was found among all the pollutants (PM_10_: *I*^2^ = 89.1, *p* < 0.001; PM_2.5_: *I*^2^ = 82.6, *p* = 0.003; NO_2_: *I*^2^ = 48.6, *p* = 0.0100; NO_x_: *I*^2^ = 91.0, *p* < 0.001), Funnel plots (Appendix A) and an Egger’s test showed publication bias only existed in research that related to PM_10_ (PM_10_: *p* = 0.018; PM_2.5_: *p* = 0.131; NO_2_: *p* = 0.156; NO_X_: N/A).

Due to the limitation of the number of included studies, we only performed a subgroup analysis of the effects of PM_10_, PM_2.5_, and NO_2_ on obesity. The results of the subgroup analysis indicated that the effects of PM_10_, PM_2.5_, and NO_2_ on obesity in children and adolescents remained significant. However, the sources of heterogeneity were not well explained in terms of the study design, study region, and study quality (Appendix A). 

The associations between exposure to PM_10_, PM_2.5_, PM_1_, and NO_2_ on childhood obesity in sensitivity analysis were generally similar and significant with the original findings (Appendix A). Meanwhile, exposure to PM_10_, PM_2.5_, and NO_2_ had relatively robust effects on childhood BMI growth (Appendix A).

## 4. Discussion

The objective of this study was to comprehensively assess the relationship between childhood exposure to air pollutants with obesity and weight status among children and adolescents. We conducted a systematic review and meta-analysis of 15 studies from 7 countries. The results showed that long-term exposure to particulate matter and NO_2_ was significantly correlated with the risk of childhood obesity, while BMI also showed similar elevated results. Although O_3_ and NO_x_ also had a positive effect on the increase in weight status, none reached significant levels. Notably, as the aerodynamic diameter of particulate matter decreases, the fattening effect on children increases, and researchers have begun to concentrate on the smaller particle size pollutants (PM_1_).

Of the 15 studies that were included in the analysis, around 73% were published in the last three years, and most were conducted in developing countries. We compared a meta-analysis of the association between air pollutants and obesity in adults [28], where half of the included studies were published after 2019, with the difference that the study areas were predominantly developed countries. An identifiable trend is that concerns about air pollution and childhood obesity are rising rapidly in developing countries. One plausible explanation is that although developed countries still maintain high rates of childhood obesity [49,50], the effect of air pollutants on body weight has been limited because air pollution levels have declined significantly in these countries [28]. Developing countries, in contrast, appear confronted with a double crisis, with increasing prevalence and growth rates of childhood obesity on the one hand [51], and deteriorating air quality resulting from urbanization and industrialization on the other [52,53]. Thus, numerous original studies were conducted in developing countries in recent years, and the effects tend to be more significant, further providing foundations for the analysis of our research.

As the main air pollutants of concern in the present study, particulate matter significantly influenced childhood obesity and BMI growth. Our findings were consistent with the previous hypothesis that with smaller aerodynamic diameters of respirable particulate matter, the more toxic compounds would be adsorbed and also more easily inhaled deep into the lungs, therefore are more harmful to health [54,55]. In contrast to children, long-term exposure to PM_10_ and PM_2.5_ showed insignificant effects on adult obesity according to a meta-analysis that was conducted on adults [28]. Similar results were found for NO_2_ and O_3_, both pollutants were positively associated with the development of obesity. As can be seen, the effects of different pollutants on people can be diverse and complex, even for the same pollutant, the impact can vary depending on the characteristics of the population (i.e., age, gender, region). These specific mechanisms require further investigation and validation.

While the mechanisms linking exposure to air pollution and obesity are not completely understood, biochemical mechanisms have been commonly mentioned and accepted as the main obesity-causing mechanisms in relation to air pollutants [12,25,56]. Firstly, from the perspective of human metabolism, air pollutants entering the body from the respiratory tract may increase oxidative stress in tissues and systems [57]. Take PM_2.5_, as an example, it can affect gene expression in mitochondria in brown adipose tissue, resulting in increased production of reactive oxygen species in brown fat stores, which lead to metabolic dysfunction [13], and susceptibility to lipid metabolism and glucose metabolism [58]. Secondly, the inflammatory response that is triggered by air pollutants can lead to vascular damage as well as insulin resistance, can also have an impact on body weight [14]. Studies also found that the occurrence of sleep-disordered breathing (SDB) was related to exposure to air pollutants [59]. Those who lived in regions with high NO_2_ and PM_2.5_ levels were much more likely to suffer from SDB, which in turn caused mental and physical health disparities [60]. Sleep deprivation correlated with decreased levels of leptin secretion, lower thyroid stimulating hormone secretion, and lower glucose tolerance, all of which may increase BMI status [15]. Finally, behavioral mechanisms can be explained in another direction [12]. For example, air pollution can reduce people’s willingness to participate in outdoor activities [16]. In addition, it can also improve the consumption of trans fats and fast food [61], which may contribute to obesity. However, attention should be drawn to the fact that while the obesogenic mechanisms of air pollution have been validated in animal models, uncertainties remain for humans or for different pollutants, and more research should be conducted to elucidate such pathways.

The strength of the present study is that it comprehensively and quantitatively assesses the relationship between long-term exposure to air pollutants and childhood obesity. Previous studies focused on the whole population [12] or adults [28], with limited studies on children and adolescents [25,62]. Meanwhile, the majority of the original studies that were included in this study were published in the past three years, and the results are relatively new. Thirdly, the exposure doses in the original studies were mostly different, thus it is difficult to compare the effects, and we converted the data in a standardized way to improve the comparability of the data. Finally, analyses and collations were performed according to the standard methods that are required by the PRISMA checklist.

However, this meta-analysis still has many limitations that need to be noted. Firstly, although we systematically searched the currently that is available epidemiological evidence, the amount of research was still limited and the potential sources of heterogeneity remain to be explored. When we integrated the results across studies, the quality of the articles varied, exposure was not measured and estimated in a standardized way, analytical methods were imperfect, and the certainty of the evidence in the articles was generally poor, so caution should be exercised in interpreting the results as well. Secondly, the findings were based on numerous cross-sectional studies; therefore, causality was difficult to be determined accurately. According to GRADE system, the level of evidence for observational studies was still low. Thirdly, due to the complexity of growth patterns, the BMI that was applied to measure obesity in adults may not be applicable to a certain extent to children and adolescents. Fourth, for younger children (before two years of age), early obesity may be associated with the subsequent catch-up growth of low birth weight due to maternal exposure to air pollution [63], and the mechanisms for infants need to be further probed. Finally, obesity is a disease with complex causes, the magnitude of the direct role that is contributed by air pollution was unclear, and residual bias (i.e., socio-economic conditions, physical activity) may still influence the outcomes. For example, high-income families (or parents with higher education levels) tend to live in more privileged residential areas with relatively lower levels of air pollution, better green spaces and have a more structured diet, while low-income families tend to be the opposite. Likewise, parents who live in more polluted areas may restrict their children’s outdoor activities to reduce the exposure to air pollutants. These potentially confounding variables were not comprehensively captured and reasonably explained in studies

The results of the study further reveal the risk of air pollution on childhood obesity. The implications of air pollutants are direct and significant, not only for human health but also for the climate. Therefore, policy-makers can also benefit from these findings that economic development and urbanization can create a number of problems, especially in developing countries, and require reflection on how to develop appropriate policies to balance economic development and environmental pollution. A synergistic approach to air pollution and climate change management that is based on global cooperation is essential. Important sectors such as transportation, energy, and manufacturing are the main focus of high emissions of PM, SO_2_, NO_x_, and GHG. It is imperative to accelerate the transformation of the energy mix and use technology to drive low-carbon production. At the same time, it is essential to establish an integrated system of atmospheric monitoring, emissions supervision, and pollution remediation.

The present review also provides insights for future studies. Firstly, long-term cohort studies with large samples of children and adolescents across age, gender, and ethnicity are required in order to provide more representative and convincing results. Secondly, the diagnosis of childhood obesity requires more diverse anthropometric measures such as waist circumference, waist-to-hip ratio, subcutaneous fat, and total and high-density lipoprotein cholesterol. In addition, the health effects of ultrafine particulate matter still need to be clarified. Finally, interactions between multiple pollutants and their effects on humans also require estimation.

## 5. Conclusions

In summary, air pollutants can be considered as a probable risk factor for the weight status of children and adolescents. Although studies are still limited, our study provides some indication. The next step is to conduct longer-term and large-scale studies on different population subgroups, exposure concentrations, and pollutant combinations to provide detailed evidence on the impacts of air pollution on human health. Measures should also be taken by the government to regulate and control the emission of air pollutants to provide a pleasant living environment for residents.

## Figures and Tables

**Figure 1 ijerph-19-04491-f001:**
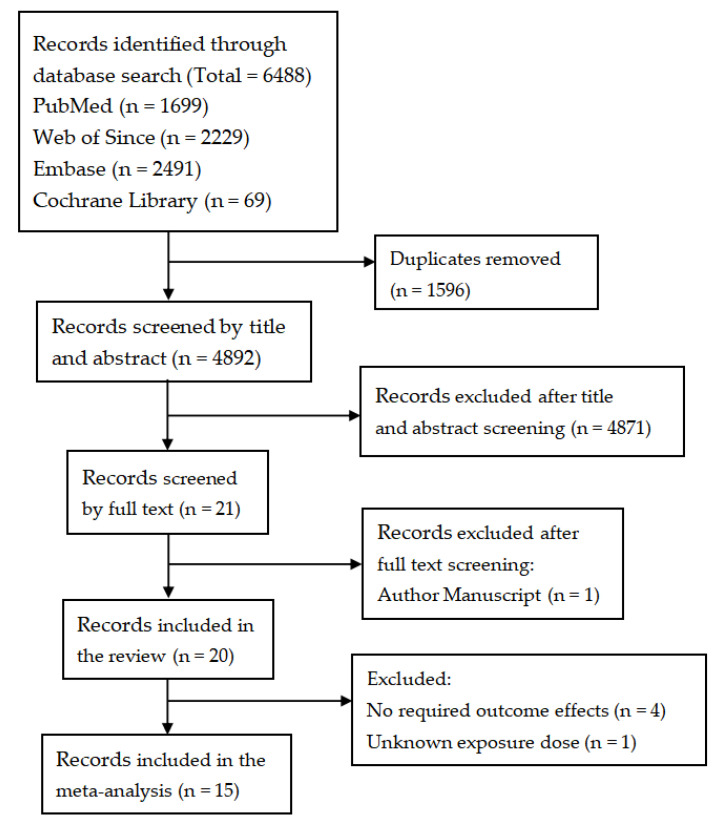
Flowchart of study selection.

**Figure 2 ijerph-19-04491-f002:**
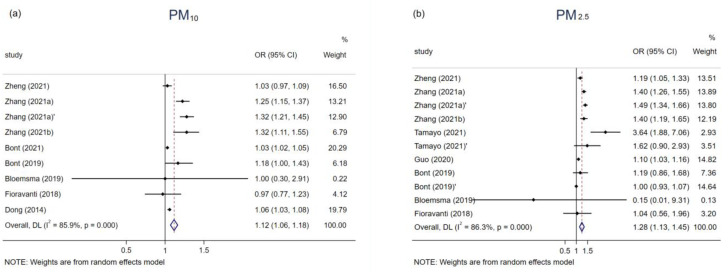
Associations of PM_10_ (**a**), PM_2.5_ (**b**), PM_1_ (**c**), O_3_ (**d**), and NO_2_ (**e**) with obesity in children and adolescents.

**Figure 3 ijerph-19-04491-f003:**
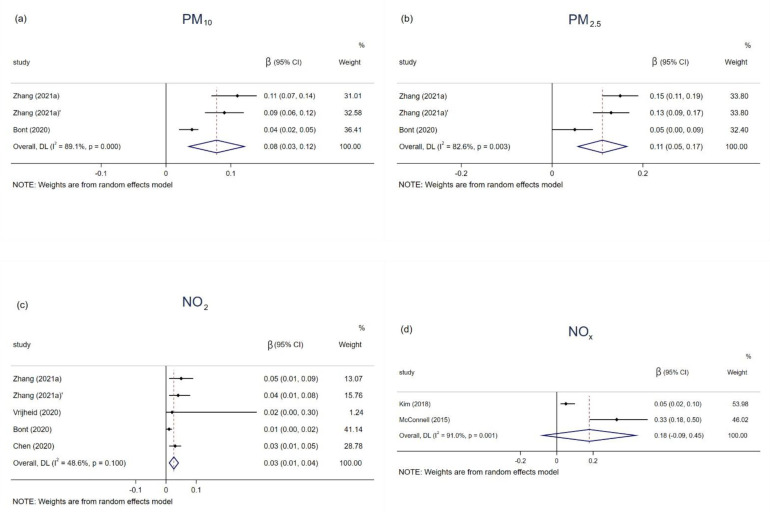
Associations of PM_10_ (**a**), PM_2.5_ (**b**), NO_2_ (**c**), and NO_x_ (**d**) with BMI in children and adolescents.

**Table 1 ijerph-19-04491-t001:** Characteristics of the 15 included studies on the association between air pollution and childhood obesity.

Study ID	Author (Year)	Country	Study Design	Study Period	Sample Size (Boy %)	Age	Quality ^a^
1	Zheng et al. (2021)	China	Cross-sectional	2019	36,456 (52.1)	9–17	13
2	Zhang et al. (2021a)	China	Cross-sectional	2013–2014	44,718 (50.5)	7–18	13
3	Zhang et al. (2021b)	China	Cross-sectional	2013–2014	9897 (50.3)	10–18	13
4	Tamayo et al. (2021)	Mexico	Cross-sectional	2006 and 2012	4306 (51.5)	2–18	11
5	Bont et al. (2021)	Spain	Cohort	2006–2018	416,955 (51.4)	2–15	12
6	Vrijheid et al. (2020)	UK	Cross-sectional	2013–2016	1301 (54.7)	6–11	11
7	Guo et al. (2020)	China	Cross-sectional	2013–2014	40,953 (48.3)	6–17	13
8	Bont et al. (2020)	Spain	Cohort	2011–2016	79,992 (51.0)	0–5	13
9	Chen et al. (2020)	China	Cohort	2012–2014	5752 (52.5)	0–2	12
10	Bont et al. (2019)	Spain	Cross-sectional	2012	2660 (51.1)	7–10	13
11	Bloemsma et al. (2019)	Netherlands	Cohort	1996–2014	3680 (51.9)	3–17	12
12	Kim et al. (2018)	US	Cohort	2002–2003	2318 (50.6)	6.5 ± 0.7	13
13	Fioravanti et al. (2018)	Italy	Cohort	2003–2004	719 (50.6)	4–8	12
14	McConnell et al. (2015)	US	Cohort	2003–2014	3318 (49.6)	10.1 ± 0.59	13
15	Dong et al. (2014)	China	Cross-sectional	2009	30,056 (50.4)	2–14	11

Abbreviations: US, United States of America; UK, United Kingdom of Great Britain and Northern Ireland. ^a^ National Institutes of Health’s Quality Assessment Tool for Observational Cohort and Cross-Sectional Studies (NIH-QAT)

**Table 2 ijerph-19-04491-t002:** Exposure, outcome, and statistical information of the 15 included studies on the association between air pollution and childhood obesity.

Study ID	Author (Year)	Exposure	Duration	Exposure Assessment	Outcome Definition	Statistical Model	Adjusted Covariates
1	Zheng et al. (2021)	PM_10_, PM_2.5_, O_3_, NO_2_	Long-term	Monitoring stations	Age-and-sex specific BMI cut-offs (Chinese national standard)	Multivariate regression model	Sex, age, paternal, sugar-sweetened beverage consumption, sweetened food consumption, frequency of having breakfast, fried food consumption, physical activity duration
2	Zhang et al. (2021a)	PM_10_, PM_2.5_, PM_1,_ NO_2_	Long-term	Satellite-based spatial-temporal model	Age-and-sex specific BMI cut-offs (Chinese national standard)	Mixed-effects linear and logistic regression models	Age, physical activity, fruit & vegetable intake, parental smoking, parental education, north or south, urban residency, regional GDP per capita
3	Zhang et al. (2021b)	PM_10,_ PM_2.5_,PM_1_, NO_2_	Long-term	Satellite-based spatial-temporal model	Waist circumference (Chinese national standard)	Generalized linear mixed-effects models	Age, sex, weight status, temperature, relative humidity, parental education level achieved, parental smoking status, parental alcohol consumption, family history of type 2 diabetes, hypertension, obesity, or cerebrovascular disease, outdoor physical activity time, diet of high fat, SSBs intake.
4	Tamayo et al. (2021)	PM_2.5_	Long-term	Hybrid spatio-temporal model	Age-specific BMI (WHO standard)	Logistic regression models	Age, sex, SES, and smoking status
5	Bont et al. (2021)	PM_10_, PM_2.5_, NO_2_	Long-term	Land use regression model	Age-and-sex specific BMI (WHO standard)	Cox proportional hazards models	Sex, deprivation index, nationality, deprivation index, and had age (1-year categories) in the strata statement.
6	Vrijheid et al. (2020)	NO_2_	Long-term	Land use regression model	Age-and-sex specific BMI (WHO standard)	Linear regression models, and logistic regression models	Sex, maternal BMI, maternal education, maternal age at conception, parity, parental country of origin, breastfeeding, and birth weight
7	Guo et al. (2020)	PM_2.5_	Long-term	Machine-learning model	Age-and-sex specific BMI cut-offs (Chinese national standard)	Logistic regression models	Sex, age, urbanity, boarding school or not, economic level, maternal occupation, maternal education, vegetable intake, fruit intake, beverages intake, activity times, ventilation, cooking fuel type, household heating fuel type, school heating fuel type, and secondhand smoke duration
8	Bont et al. (2020)	PM_10_, PM_2.5_, NO_2_	Long-term	Land use regression model	BMI z-scores (WHO standard)	Linear spline multilevel model	Sex, age, deprivation index, nationality
9	Chen et al. (2020)	NO_2_	Long-term	Land use regression model	Age- and sex-specific z scores for BMI (WHO standard)	Generalized estimating equation models, Distributed lag nonlinear models	Maternal age, maternal education, annual household income and residence area
10	Bont et al. (2019)	PM_10_, PM_2.5_, NO_2_	Long-term	Land use regression model	Age- and sex-specific z scores for BMI (WHO standard)	Multilevel mixed linear and ordered logistic models	Maternal and paternal education, maternal and paternal country of birth, paternal employment status, number of siblings, household status and maternal smoking during pregnancy
11	Bloemsma et al. (2019)	PM_10_, PM_2.5_, NO_2_	Long-term	Land use regression model	Age-and-sex specific BMI (International Obesity Task Force cut-offs)	Generalized linear mixed models	Age, sex maternal level of education, paternal level of education, maternal smoking during pregnancy, parental smoking in child’s home and neighborhood socioeconomic status and region
12	Kim et al. (2018)	NO_x_	Long-term	California line-source dispersion model	BMI (US CDC criteria)	Linear mixed effects models	Age, sex, race/ethnicity, parental education, and Spanish baseline questionnaire
13	Fioravanti et al. (2018)	PM_10_, PM_2.5_, NO_2_	Long-term	Land use regression model	Age- and sex-specific z scores for BMI (WHO standard)	Logistic regression models, Generalized Estimating Equation models and linear regression models	Maternal and paternal education, maternal pre-pregnancy BMI, maternal smoking during pregnancy, gestational diabetes, maternal age at delivery, gestational age, childbirth weight, breastfeeding duration, age at weaning and inversely weighted for the probability of participation at baseline and at the two follow-ups, respectively
14	McConnell et al. (2015)	NO_x_	Long-term	California line-source dispersion model	Age-and-sex specific BMI (US CDC criteria)	Multilevel linear model	Sex, ethnicity, community, year of enrollment, and age
15	Dong et al. (2014)	PM_10_, NO_2_, SO_2_, O_3_	Long-term	Monitoring stations	Age-and-sex specific BMI standards (Chinese CDC criteria)	Logistic regression	Age, gender, parental education, breastfeeding, low birth weight, area of residence per person, house decorations, home coal use, ventilation device in kitchen, air exchange in winter, passive smoking exposure, and districts

Abbreviations: PM_10_, particulate matter with the diameter ≤ 10 mm; PM_2.5_, particulate matter with diameter ≤ 2.5 mm; PM_1_, particulate matter with the diameter ≤ 1 mm; NO_2_, nitrogen dioxide; NO_x_, nitrogen oxides; SO_2_, sulfur dioxide; O_3_, ozone; BMI, body mass index; WHO, World Health Organization; US, The United States; CDC, Center for Disease Control and Prevention.

**Table 3 ijerph-19-04491-t003:** Summary effects and 95% confidence intervals of each pollutant on obesity and BMI in children and adolescents.

Pollution Type	Author (Year)	Group	Sample Size	Incremental Scale	Original Effect	Transformed OR/β
**obesity**						
PM_10_	Zheng (2021)	Total	36,456	10 μg/m^3^	1.03 (0.97, 1.09)	-
	Zhang (2021a)	Boy	22,573	10 μg/m^3^	1.25 (1.15, 1.37)	-
	Zhang (2021a)’	Girl	22,145	10 μg/m^3^	1.32 (1.21, 1.45)	-
	Zhang (2021b)	Total	44,718	10 μg/m^3^	1.32 (1.11, 1.55)	-
	Bont (2021)	Total	416,955	6.4 μg/m^3^	1.02 (1.02, 1.03)	1.03 (1.02, 1.05)
	Bont (2019)	Home	2660	5.6 μg/m^3^	1.10 (1.00, 1.22)	1.18 (1.00, 1.43)
	Bloemsma (2019)	Total	3680	1.06 μg/m^3^	1.00 (0.88, 1.12)	1.00 (0.30, 2.91)
	Fioravanti (2018)	Total	719	10 μg/m^3^	0.97 (0.77, 1.23)	-
	Dong (2014)	Total	30,056	31 µg/m^3^	1.19 (1.11, 1.26)	1.06 (1.03, 1.08)
PM_2.5_	Zheng (2021)	Total	36,456	10 μg/m^3^	1.19 (1.05, 1.33)	-
	Zhang (2021a)	Boy	22,573	10 μg/m^3^	1.40 (1.26, 1.55)	-
	Zhang (2021a)’	Girl	22,145	10 μg/m^3^	1.49 (1.34, 1.66)	-
	Zhang (2021b)	Total	44,718	10 μg/m^3^	1.40 (1.19, 1.65)	-
	Tamayo (2021)	Children	1370	10 μg/m^3^	3.64 (1.88, 7.06)	-
	Tamayo (2021)’	Adolescence	1519	10 μg/m^3^	1.62 (0.90, 2.93)	-
	Guo (2020)	Total	40,953	10 μg/m^3^	1.10 (1.03, 1.16)	-
	Bont (2019)	Home	2660	2.7 μg/m^3^	1.05 (0.96, 1.15)	1.19 (0.86, 1.68)
	Bont (2019)’	School	2660	10.7 μg/m^3^	1.00 (0.93, 1.08)	1.00 (0.93, 1.07)
	Bloemsma (2019)	Total	3680	1.17 μg/m^3^	0.80 (0.59 1.09)	0.15 (0.01, 9.31)
	Fioravanti (2018)	Total	719	5 μg/m^3^	1.02 (0.75, 1.40)	1.04 (0.56, 1.96)
PM_1_	Zhang (2021a)	Boy	22,573	10 μg/m^3^	1.38 (1.21, 1.57)	-
	Zhang (2021a)’	Girl	22,145	10 μg/m^3^	1.44 (1.25, 1.67)	-
	Zhang (2021b)	Total	44,718	10 μg/m^3^	1.42 (1.23, 1.64)	-
O_3_	Zheng (2021)	Total	36,456	10 μg/m^3^	1.04 (1.00, 1.08)	-
	Dong (2014)	Total	30,056	11.3 ppb	1.14 (1.04, 1.24)	1.06 (1.02, 1.09)
NO_2_	Zheng (2021)	Total	36,456	10 μg/m^3^	1.13 (1.04, 1.22)	-
	Zhang (2021a)	Boy	22,573	10 μg/m^3^	1.14 (1.04, 1.24)	-
	Zhang (2021a)’	Girl	22,145	10 μg/m^3^	1.21 (1.10, 1.34)	-
	Zhang (2021b)	Total	44,718	10 μg/m^3^	1.44 (1.22, 1.71)	-
	Bont (2021)	Total	416,955	21.8 μg/m^3^	1.03 (1.02, 1.04)	1.01 (1.00, 1.02)
	Chen (2020)	Total	5752	10 μg/m^3^	1.11 (1.00, 1.22)	-
	Bont (2019)	Home	2660	13.7 μg/m^3^	1.05 (0.97, 1.13)	1.04 (0.98, 1.09)
	Bont (2019)’	School	2660	22.3 μg/m^3^	1.09 (0.92, 1.28)	1.04 (0.96, 1.12)
	Bloemsma (2019)	Total	3680	8.9 μg/m^3^	1.40 (1.12, 1.74)	1.46 (1.14, 1.86)
	Fioravanti (2018)	Total	719	10 μg/m^3^	0.99 (0.86, 1.12)	-
	Dong (2014)	Total	300,56	5.3 ppb	1.13 (1.04, 1.22)	1.13 (1.04, 1.21)
**BMI**						
PM_10_	Zhang (2021a)	Boy	22,573	10 μg/m^3^	0.11 (0.07, 0.14)	-
	Zhang (2021a)’	Girl	22,145	10 μg/m^3^	0.09 (0.06, 0.12)	-
	Bont (2020)	Total	79,992	6.3 μg/m^3^	0.02 (0.01, 0.03)	0.04 (0.02, 0.05)
PM_2.5_	Zhang (2021a)	Boy	22,573	10 μg/m^3^	0.15 (0.11, 0.19)	-
	Zhang (2021a)’	Girl	22,145	10 μg/m^3^	0.13 (0.09, 0.17)	-
	Bont (2020)	Total	79,992	1.5 μg/m^3^	0.01 (0.00, 0.01)	0.05 (0.00, 0.09)
NO_2_	Zhang (2021a)	Boy	22,573	10 μg/m^3^	0.05 (0.01, 0.09)	-
	Zhang (2021a)’	Girl	22,145	10 μg/m^3^	0.04 (0.01, 0.08)	-
	Vrijheid (2020)	Total	1301	92.8 μg/m^3^	0.15 (0.01, 0.28)	0.02 (0.00, 0.30)
	Bont (2020)	Total	79,992	21.3 μg/m^3^	0.02 (0.01, 0.03)	0.01 (0.00, 0.02)
	Chen (2020)	Total	5752	10 μg/m^3^	0.03 (0.01, 0.05)	-
NO_x_	Kim (2018)	Total	2318	9.4 ppb	0.10 (0.03, 0.20)	0.05 (0.02, 0.10)
	McConnell (2015)	Total	2994	16.8 ppb	1.13 (0.61, 1.65)	0.33 (0.18, 0.50)

Abbreviations: PM_10_, particulate matter with the diameter ≤ 10 mm; PM_2.5_, particulate matter with diameter ≤ 2.5 mm; PM_1_, particulate matter with the diameter ≤ 1 mm; NO_2_, nitrogen dioxide; NO_x_, nitrogen oxides; O_3_, ozone; BMI, body mass index; OR, odds ratio; β, regression coefficient.

## Data Availability

Not applicable.

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
