# Peer review of "The Association between Childhood Exposure to Ambient Air Pollution and Obesity: A Systematic Review and Meta-Analysis"

_ijerph, 2022, doi:10.3390/ijerph19084491_

Round 1
Reviewer 1 Report
The study proposed by the authors is interesting. It was carried out according to modern standards and good practices for conducting and validating a meta-analysis. However, I still have some suggestions.
It will be necessary to choose a single statistical model a priori, I think for your case the random effect or mixed effect model, and to carry out all the aggregations. Fitting the models a posteriori predisposes to a certain estimation bias and ill-fitting discussions.
For the publication bias, I advise you to add the funnel plots to get an idea of the extent of the bias.
For the discussion, given the low quality of some of the included studies, develop even more the causal links to better alert the readers on the external validity of the result of the meta-analysis. you started well "Finally, obesity as a disease with complex causes, the magnitude of the direct role contributed by air pollution was unclear, and residual bias (i.e., socio-economic conditions, physical activity) may still influence the outcomes." so develop more these aspects.
Author Response
Dear Editors and Reviewers,
Thanks very much for taking your time to review this manuscript. We really appreciate all your generous comments and suggestions! Please find my itemized responses in below and my revisions in the re-submitted files. We have also revised the English expressions in the article. If there are any other modifications we could make, we would like very much to modify them and we really appreciate your help.
1. It will be necessary to choose a single statistical model a priori, I think for your case the random effect or mixed effect model, and to carry out all the aggregations. Fitting the models a posteriori predisposes to a certain estimation bias and ill-fitting discussions.
Response 1:According to your comments, and given the general heterogeneity of the studies we included, we have uniformly chosen random effects model for conservative estimates.
2. For the publication bias, I advise you to add the funnel plots to get an idea of the extent of the bias.
Response 2:For publication bias, we supplemented the funnel plot (Supplementary Figure 1 & Supplementary Figure 2) by observing the funnel plot distribution along with the results of the Egger's test (Supplementary Table 2) to assess publication bias.
3. For the discussion, given the low quality of some of the included studies, develop even more the causal links to better alert the readers on the external validity of the result of the meta-analysis.
Response 3:We conducted subgroup analyses based on scores of study quality, and further suggested in the discussion the need for caution in the interpretation of results. Please see lines 304-310: “Firstly, although we systematically searched the currently available epidemiological evidence, the amount of research was still limited and the potential sources of heterogeneity remain to be explored. When we integrated the results across studies, the quality of the articles varied, exposure was not measured and estimated in a standardized way, analytical methods were imperfect, and the certainty of the evidence in the articles was generally poor, so caution should be exercised in interpreting the results as well.”
4. you started well "Finally, obesity as a disease with complex causes, the magnitude of the direct role contributed by air pollution was unclear, and residual bias (i.e., socio-economic conditions, physical activity) may still influence the outcomes." so develop more these aspects.
Response 4:We further illustrated and discussed the potential confounding factors. Please see lines 318-326: “Finally, obesity as a disease with complex causes, the magnitude of the direct role contributed by air pollution was unclear , and residual bias (i.e., socio-economic conditions, physical activity) may still influence the outcomes. For example, high-income families (or parents with higher education levels) tend to live in more privileged residential areas with relatively lower levels of air pollution, better green spaces and have a more structured diet, while low-income families tend to be the opposite. Likewise, parents who live in more polluted areas may restrict their children's outdoor activities to reduce the exposure to air pollutants. These potentially confounding variables were not comprehensively captured and reasonably explained in studies.”

Reviewer 2 Report
The study is very interesting and of high importance. It is a very important topic to tackle, especially with the growing impact of climate change and increasing rates of overweight and obesity among children and adolescents worldwide. Findings can be used by policymakers to reduce air pollutants and improve the quality of air.
The study design and analysis are very robust. The methodology is well described, and the manuscript is well written. Minor comments ought to be taken into consideration:
- Add more details to the titles of figures and tables. They are quite general and should at least refer to the study population. Also, there is a typo in the title of table S3.
- Discussion: the first sentence is also quite general, please add the study population “among children and adolescents”.
- Discussion: it is very well written; however, it is worth mentioning the effects of air pollutants on climate change very briefly so the study can be related to global concerns as well. Findings of this study could be used to establish and reinforce policies and actions to end climate change.
- Table S1: Question 5: how did study calculate their sample size?
- Table S1: Question 6, 10, 13: how were these questions assessed for cross-sectional studies?
Author Response
Dear Editors and Reviewers,
Thanks very much for taking your time to review this manuscript. We really appreciate all your generous comments and suggestions! Please find my itemized responses in below and my revisions in the re-submitted files. We have also revised the English expressions in the article. If there are any other modifications we could make, we would like very much to modify them and we really appreciate your help.
1. Add more details to the titles of figures and tables. They are quite general and should at least refer to the study population. Also, there is a typo in the title of table S3.
Response 1: We have revised the titles of all figures and tables and supplemented the descriptions of the study population and study content. The spelling of "subgroup" in Supplementary Table 3 has been modified.
2. Discussion: the first sentence is also quite general, please add the study population “among children and adolescents”.
Response 2: In the first sentence of the discussion, the subjects of the study were added: children and adolescents.Please see lines 239-241.
3. Discussion: it is very well written; however, it is worth mentioning the effects of air pollutants on climate change very briefly so the study can be related to global concerns as well. Findings of this study could be used to establish and reinforce policies and actions to end climate change.
Response 3: We complemented by insights and discussions relevant to policy decisions and environmental governance. Please see lines 327-338: “The results of the study further reveal the risk of air pollution on childhood obesity. The implications of air pollutants are direct and significant, not only for human health but also for the climate. Therefore, policy makers can also benefit from these findings that economic development and urbanization can create a number of problems, especially in developing countries, and require reflection on how to develop appropriate policies to balance economic development and environmental pollution. A synergistic approach to air pollution and climate change management based on global cooperation is essential. Important sectors such as transportation, energy and manufacturing are the main focus of high emissions of PM, SO2, NOx, and GHG. It is imperative to accelerate the transformation of the energy mix and use technology to drive low-carbon production. At the same time, establish an integrated system of atmospheric monitoring, emissions supervision and pollution remediation.”
4.Table S1: Question 5: how did study calculate their sample size?
Response 4: In all 15 included studies, the data were based on cohort or cross-sectional studies that had been conducted in each country or region. Although the source of the data and the sample size were mentioned in the articles, no studies specified how the sample size was calculated to obtain it, and no sample size justification, power description, or variance and effect estimates were performed. Therefore, according to the scoring principles of the scale we assigned a score of 0 to this item for all articles.
5. Table S1: Question 6, 10, 13: how were these questions assessed for cross-sectional studies?
Response 5: With regard to the exposure period (Q6), if historical data were used to assess air pollution, then we considered the exposure measurement to be earlier than the outcome measurement, and assign a score of 1.There is also the case where recent air pollution levels were used to predict long-term exposure, which we then assign a score of 0.
Q10 is about the measurement times of exposure, since air pollution is a long-term problem, all studies used multiple measurements or long periods of monitoring, the researchers stated in the original article how and when the air pollutants were collected, so we assumed that exposure was measured multiple times, all assigned a score of 1.
Q13, regarding the loss ratio of follow-up, due to the characteristics of cross-sectional studies, we uniformly considered the lost rate to be < 20%, and assigned a value of 1.

Round 2
Reviewer 1 Report
The authors have adequately answered the questions.